# Yield and Rhizosphere Soil Environment of Greenhouse Zucchini in Response to Different Planting and Breeding Waste Composts

**DOI:** 10.3390/microorganisms11041026

**Published:** 2023-04-14

**Authors:** Jianzhong Tie, Yali Qiao, Ning Jin, Xueqin Gao, Yayu Liu, Jian Lyu, Guobin Zhang, Linli Hu, Jihua Yu

**Affiliations:** 1College of Horticulture, Gansu Agricultural University, Lanzhou 730070, China; tiejianzhong1999@163.com (J.T.);; 2Gansu Provincial Key Laboratory of Arid Land Crop Science, Gansu Agricultural University, Lanzhou 730070, China

**Keywords:** planting and breeding waste, compost, zucchini, soil environment

## Abstract

Composting, planting, and breeding waste for return to the field is the most crucial soil improvement method under the resource utilization of agricultural waste. However, how the vegetable yield and rhizosphere soil environment respond to different composts is still unknown. Therefore, eight formulations were designed for compost fermentation using agricultural waste [sheep manure (SM), tail vegetable (TV), cow manure (CM), mushroom residue (MR), and corn straw (CS)] without fertilizer (CK1) and local commercial organic fertilizer (CK2) as controls to study the yield and rhizosphere soil environment of greenhouse zucchini in response to different planting and breeding waste compost. Applying planting and breeding waste compost significantly increased the soil’s organic matter and nutrient content. It inhibited soil acidification, which T4 (SM:TV:CS = 6:3:1) and T7 (SM:TV:MR:CS = 6:2:1:1) treatments affected significantly. Compared to CK2 treatment, T4 and T7 treatments showed a greater increase, with a significant increase of 14.69% and 11.01%, respectively. Therefore, T4, T7, and two control treatments were selected for high-throughput sequencing based on yield performance. Compared with the CK1 treatment, although multiple applications of chemical fertilizers led to a decrease in bacterial and fungal richness, planting and breeding waste compost maintained bacterial diversity and enhanced fungal diversity. Compared to CK2, the relative abundance increased in T7-treated *Proteobacteria* (*Sphingomonas*, *Pseudomonas*, and *Lysobacter*) and T4-treated *Bacteroidetes* (*Flavobacterium*) among bacteria. An increase in T4-treated *Ascomycota* (*Zopfiella* and *Fusarium*) and *Basidiomycota* among fungi and a decrease in T7-treated *Mortierellomycota* have been observed. Functional predictions of the bacterial Tax4Fun and fungal FUNGuild revealed that applying planting and breeding waste compost from the T4 treatment significantly increased the abundance of soil bacterial Metabolism of Cities, Genetic Information Processing, and Cellular Processes decreased the abundance of Pathotroph and Saprotroph-Symbiotroph fungi and increased the abundance of Saprotroph fungi. Overall, planting and breeding waste compost increased zucchini yield by improving soil fertility and microbial community structure. Among them, T4 treatment has the most significant effect, so T4 treatment can be selected as the optimized formulation of local commercial organic fertilizer. These findings have valuable implications for sustainable agricultural development.

## 1. Introduction

The sustainable utilization of agricultural wastes continues to receive global attention as it aims to reduce the negative environmental impact of waste and can contribute to proper waste-handling processes [1]. China is the world’s largest producer of agricultural waste. Agricultural waste is dominated by planting and breeding waste. According to statistics, China has 3.8 billion tons of breeding waste (livestock and poultry manure) each year, and the comprehensive utilization rate is less than 60%. During the storage and disposal of animal waste, a range of pollutants, including nutrients, pathogens, and heavy metals, entered local farmland soils, surface water, and groundwater, which pose direct and indirect human health risks [2]. The annual production of straw is nearly 900 million tons; about 200 million tons are unused, and about 260 million tons of vegetable waste are generated, which seriously affect the sustainable development of agriculture [3]. At present, due to the excessive application of chemical fertilizers in China’s agricultural production and the high intensity use of arable land, the problem of arable land quality has gradually appeared. The fertilizer utilization of waste can bring back the nutrient resources in a large amount of waste from the farming industry to the soil, solving the contradiction of “Use land to raise land”. It can also form a waste resource utilization model with other key technologies to promote the safe, efficient, and recycling use of agricultural waste and reduce environmental pressure [4]. The composting of planting and breeding waste for return to the field is the most crucial method of soil improvement under the category of resource utilization of agricultural waste. Composting is the process of reducing pollution through the consumption of breeding waste, and many harmful substances are decomposed and used by microorganisms in this process. In addition, the organic fertilizer made by composting is rich in nutrients, has high microbial activity, and has a unique structural composition that allows for better improvement of the soil’s physical, chemical, and biological properties, which are not available in chemical fertilizers. Therefore, organic compost is widely used in soil nutrient and ecological environment improvement [5]. Murat Durmuş and R. Kızılkaya [6] have composted tomato waste for greenhouse cultivation; after applying compost, tomato yield increased, and the biological properties of the soil improved. Research showed that, when combined with organic fertilizers such as spent mushroom substrate compost, beneficial microbes could promote plant growth and yield as well as suppress plant disease by sustaining soil fertility through complex bacteria–soil–plant interactions [7]. The use of planting and breeding waste for compost can realize the effective transformation of biological resources, increase the value of resource utilization, and reduce environmental pollution while expanding the range of raw materials for composting, thereby reducing costs and turning waste into value.

The application of organic fertilizers increased soil organic matter, soil porosity, and water-holding capacity, thus improving soil quality [8]. Lucas et al. [9] showed that one of the reasons why the use of organic fertilizers regulates soil microbial community structure and promotes soil aggregation is that the cellulose in organic fertilizers contains a large amount of bioavailable carbon, which promotes fungal proliferation, improves soil structure, and stabilizes soil aggregates. Soil microorganisms play a crucial role in energy flow and nutrient cycling in soil and participate in the decomposition of organic matter, the degradation of xenobiotics, soil carbon sequestration, and the prevention of crop diseases [10,11]. Soil enzymes play an essential role in regulating soil nutrient turnover. These enzymes are involved in transforming soil C, N, and P, such as sucrase, urease, and alkaline phosphatase [12]. Using compost can improve soil enzyme activity and promote nutrient cycling. Soil environmental factors and soil microorganisms interact and complement each other, jointly affecting arable land productivity and crop growth [13]. Fertilizer application can indirectly cause changes in microbial growth, metabolic processes, and community composition by changing soil physicochemical properties and nutrient content. In contrast, changes in soil nutrients, as metabolic substrates for soil enzymes, also affect the activity of soil enzymes. On the other hand, soil microorganisms will participate in the decomposition and transformation of soil organic matter. Changes in their composition also lead to changes in their metabolism, mortality, and the soil’s physicochemical characteristics. Some studies have found that food waste compost could be an alternative to chemical fertilizer to increase soil microbial populations and enzyme activities and promote soil nutrients for lettuce growth [14]. Sato et al. [15] have confirmed that the application of animal waste compost reduced Cd uptake in spinach, and cattle compost with a high affinity for Cd and low P content should be the preferred soil amendment. Other studies have found that agricultural waste-based composts exhibit suppressive effects on diseases caused by the phytopathogenic soil-borne fungi Rhizoctonia solani and Sclerotinia minor [16].

Previous studies have focused on the effects of chemical and organic fertilizers alone or in combination on yield and quality of crops as well as the soil environment [14,17]. However, few studies have investigated the yield and rhizosphere soil environment in response to different planting and breeding waste composts. Moreover, the changes in crop yield, soil bacterial and fungal community diversity, and the mechanisms of interaction between compost application systems remain unclear. As discussed above, excessive and unreasonable disposal of planting and breeding waste has adverse effects on the environment, whereas compost has a beneficial impact on crop growth and soil remediation. To explore how the crops and soil environment respond to the agricultural fertilization system would be conducive to the development of sustainable agriculture. Lanzhou city, Gansu province, is a representative area of Northwest China’s agro-pastoral interlacing zone, which has rich resources of farming waste. Currently, the primary local organic fertilizer utilization method is to mix sheep manure and highland summer vegetable waste at a weight ratio of 6.5:3.5 and then aerobically ferment it into organic fertilizer. Still, the fertilizer recycling mode needs to deal with other farming waste effectively. Therefore, the experiment was conducted to study the zucchini yield and rhizosphere soil environment in response to different planting and breeding composts, which were aerobically composted according to designed compost formulations based on local compost formulations with the main local planting and breeding waste (SM, TV, CS, CM, MR) as the raw material. The aim of our study is to optimize the local compost recipe and composting treatment system for planting and breeding waste, consume part of other planting and breeding wastes, and apply compost products for crop yield, quality improvement, as well as soil improvement to realize the “production-use” combination. Ultimately, it provides a basis for the resource utilization of agricultural waste in agricultural pastoral areas of the northwest of China.

## 2. Materials and Methods

### 2.1. Site and Soil Description

The experimental site was one of interlocking agricultural and pastoral areas in Lanzhou City, Gansu Province, China (35°87′ N, 104°23′ E), from September 2021 to March 2022. The region has a temperate semi-arid continental climate with distinct seasons, a mean altitude of 1790 m above sea level, a mean annual temperature of 6.6 ℃, a mean annual rainfall of 300–400 mm, a mean annual evaporation of 1343 mm, and a frost-free period of approximately 150 days. The test field had gentle terrain, and the fertility level was uniform. The sunroom used in the experiment was cultivated for three years; the previous crop was tomato. The test soil was yellow cotton soil, rich in calcium carbonate, and the main physicochemical parameters of the 0–20 cm topsoil were fast-acting phosphorus (131 mg·kg^−1^), fast-acting potassium (376 mg·kg^−1^), organic matter (5.31 g·kg^−1^), electrical conductivity (296 μS·cm^−1^), and pH 8.15.

### 2.2. Study Materials and Compost Fermentation

The Zucchini (‘Dong Feng No. 5’ variety) was used as the test material. The conventional fertilizers used in this study were urea (N ≥ 46%), calcium superphosphate (P_2_O_5_ ≥ 16%), and potassium sulfate (K_2_O ≥ 52%). The composting fermenting bacterium used in the experiment (cellulase activity ≥60 μ/g, protease activity ≥70 μ/g, moisture content ≤20%, and colony-forming units (CFU) ≥500 hundred million·g^−1^) was purchased from Hebi City Renyuan Biological Technology Development Co., Ltd. (Hebi, China) 

The raw materials for compost fermentation (sheep mature, SM; tail vegetables, TV; cow mature, CM; mushroom residue, MR; corn straw, CS) are mixed according to the design formula, and 1 kg of composting fermenting bacteria is added for every 15~18 t. The moisture content of the pile is adjusted to 60~65%. Pile onto a platform with a height of 1 m, a bottom width of 2 m, a top width of 1 m, and an unlimited length (that is, the section is trapezoidal). The temperature was monitored 25 cm away from the top of the stack. When the temperature exceeded 60 ℃, the stack was turned for the first time and mixed every seven days. When the pile temperature was close to the ambient temperature, the color turned brown, and the grass smelled slightly. The fermentation was completed in about 38–40 days. The basic physical and chemical properties of the test planting and breeding waste materials were shown in Appendix A. The basic physical and chemical properties of different planting and breeding waste composts were presented in Appendix A.

### 2.3. Experimental Design 

The field experiment was performed in a completely randomized block design with three replicates. All treatments were as follows: CK1, no fertilizer; CK2, SM:TV = 6.5:3.5 (local commodity composting formula); T1, SM:TV = 5.5:4.5; T2, SM:TV:CM = 6:3:1; T3, SM:TV:MR = 6:3:1; T4, SM:TV:CS = 6:3:1; T5, SM:TV:CM:MR = 6:2:1; T6, SM:TV:CM:CS = 6:2:1:1; T7, SM:TV:MR:CS = 6:2:1:1 (weight ratios). The ridge, double row, and black-film covered cultivation modes were used in the experiments. Each treatment included three replicates, and the area of each experimental plot was 43.2 m^2^. The cultivation density was 18,890 plants ha^−1^, the ridge width was 80 cm, the ridge surface width was 60 cm, the furrow width was 40 cm, the plant spacing was 100 cm, and the row spacing was 20 cm. Before planting, the compost was applied as the base fertilizer according to the local organic fertilizer application rate of 6000 kg ha^−1^. The base fertilizer was spread and turned 25–30 cm deep to level and raise the ridge. Chemical fertilizer was applied as a catch-up fertilizer, and fertilizer dosage was consistent across treatments throughout the field fertility period (N:168.22 kg ha^−1^, P_2_O_5_:135.83 kg ha^−1^, and K_2_O:166.27 kg ha^−1^). During the growth of zucchini, the first melon should be 250–300 g for a timely harvest; the melon should not exceed 400 g. We ensured that all field management practices (other than fertilization) were consistent across treatments.

### 2.4. Yield Determination

Five plants of each replicate “S” marker were harvested, and the total yield of each marker plant was recorded from the beginning of the fruiting period (after the first melon was removed). 

### 2.5. Soil Sampling and Analysis of Soil Chemical Properties

After the end of the harvesting period, in March 2022, soil samples were collected from the rhizosphere soil of the marking plant (15–20 cm) of each replicate plot for each treatment using a stainless-steel auger. After mixing the samples for each treatment, the stones, roots, film debris, and other sundries were removed, leaving about 3 kg of pure soil as samples. The soil samples were sieved using a 2-mm mesh sieve and divided into two subsamples before thorough homogenization. One part was air-dried to determine the physical and chemical properties of the soil, and the other was stored at −80 ℃ for DNA extraction. 

The soil-water suspension (1:5 wt/vol) was shaken for 30 min and then filtered, and the pH and electrical conductivity (EC) of the filtrate were measured [17]. The pH of the filtrate was determined using a glass electrode (PHS-3E, Shanghai Jingke, Shanghai, China), and the EC was measured by inserting a conductivity meter (DSJ-308A, Shanghai Jingke) into the filtrate. The SOM content was measured using a titration method after oxidation with K_2_Cr_2_O_7_. The soil’s total nitrogen (TN), total phosphorus (TP), and total potassium (TK) were determined using the H_2_SO_4_-H_2_O_2_ wet digestion method. The TN was measured using the Kjeldahl method using a fully automatic Kjeldahl K1100F apparatus (Jinan Hanon Instruments Company, Jinan, China). The TP was measured using the Mo-Sb colorimetric method and analyzed using a UV-1780 spectrophotometer [Shimadzu Instruments (Suzhou) Co., Ltd., Suzhou, China]. The TK was analyzed using a ZEEnit 700P atomic absorption spectrometer (Analytik Jena, Jena, Germany). All samples were tested in triplicate. Soil enzyme (catalase, sucrase, urease, and alkaline phosphatase) activity was estimated by colorimetry [18]. 

### 2.6. Soil DNA Extraction and PCR Amplification

The genomic DNA of the soil microbial community was extracted from 0.5 g of soil samples using the HiPure Soil DNA Kits (Magen, Guangzhou, China) according to the manufacturer’s instructions. The DNA extract was analyzed on a 1% agarose gel, and DNA concentration and purity were determined using a NanoDrop ND-2000 UV-VIS spectrophotometer (Thermo Scientific, Wilmington, DE, USA). The hypervariable V3–V4 region of the bacterial 16S rRNA gene was amplified with the primer pairs 341F (5’-CCTACGGGNGGCWGCAG-3’) and 806R (5’-GGACTACHVGGGTATCTAAT-3’) [19], using an ABI StepOnePlus Real-Time PCR System (Life Technologies, Foster City, CA, USA). The internal transcribed spacer (ITS2) region of fungi was amplified using the primers ITS3_KYO2 (5’-GATGAAGAACGYAGYRAA-3’) and ITS4 (5’-TCCTCCGCTTATTGATATGC-3’) [20] on the same thermal cycler. The 16S rDNA target region of the ribosomal RNA gene was amplified by PCR (95 °C for 5 min, followed by 30 cycles at 95 °C for 1 min, 60 °C for 1 min, and 72 °C for 1 min, with a final extension at 72 °C for 7 min) using primers listed in the previous paragraph. A 50 μL sample of mixture contains 10 μL of 5 × Q5@ Reaction Buffer, 10 μL of 5 × Q5@ High GC Enhancer, 1.5 μL of 2.5 mM dNTPs, 1.5 μL of each primer (10 μM), 0.2 μL of Q5@ High-Fidelity DNA Polymerase, and 50 ng of template DNA. Related PCR reagents were from New England Biolabs, USA. Amplicons were extracted from 2% agarose gels, purified using the AxyPrep DNA Gel Extraction Kit (Axygen Biosciences, Union City, CA, USA) according to the manufacturer’s instructions, and quantified using the ABI StepOnePlus Real-Time PCR System (Life Technologies, Foster City, CA, USA). 

### 2.7. Illumina Miseq Sequencing and Sequence Processing

Purified amplicons were pooled in equimolar amounts and paired-end sequenced (PE250) on an Illumina platform according to the standard protocols. DNA and RNA library sequencing was performed on the Illumina HiseqTM 2500/4000 by Gene Denovo Biotechnology Co., Ltd. (Guangzhou, China). The raw sequences were analyzed using FASTP (version 0.18.0) [21] for quality control to get high-quality clean reads. Paired-end clean reads were merged as raw tags using FLASH [22] (version 1.2.11) with a minimum overlap of 10 bp and mismatch error rates of 2%. The reads were replicated, sorted, and clustered into operational taxonomic units (OTUs) at the default 97% similarity using UPARSE (version 9.2.64) [23]. All chimeric tags were removed using the UCHIME algorithm [24] and we finally obtained effective tags for further analysis. Each cluster selected the tag sequence with the highest abundance as a representative sequence. The representative OTU or ASV sequences were classified into organisms by a naïve Bayesian model using the RDP classifier [25] (version 2.2) based on the SILVA database [26] (version 138.1), the UNITE database [27] (version 8.3), or the ITS2 database [28] (version update_2015), with a confidence threshold value of 0.8.

### 2.8. Statistical Analysis

Microsoft Excel 2013 was used to calculate the mean and standard error (SE) of the crop yield and quality of zucchini and the chemical properties of soil in different treatment groups. A one-way analysis of variance (ANOVA) was applied to evaluate the effects of other treatments on the zucchini yield and soil chemical properties. Duncan’s multiple range test (significance level: *p* < 0.05) was used to compare the differences in mean values among different fertilization treatments. The SPSS software (version 23.0, IBM Corp., NY, United States) was used for one-way ANOVA.

Mothur (version v.1.30, http://www.mothur.org/) was used to analyze the diversity of microbial communities, accessed on 20 April 2022. The abundance statistics of each taxonomy were visualized using Krona [29] (version 2.6). The stacked bar plot of the community composition was visualized in the R project’s ggplot2 package [30] (version 2.2.1). Pearson correlation analysis of species was calculated in the R project psych package [31] (version 1.8.4). Chao1, ACE, Shannon, and Simpson were calculated in QIIME [32] (version 1). To analyze beta diversity, we used unweighted UniFrac distances to generate principal coordinate analysis (PCoA) maps to assess the differences between the bacterial community members of soil samples with different fertilization treatments [33]. In addition, we used PCoA based on Bray–Curtis distances to evaluate differences between the fungal community members of soil samples with different fertilization treatments.

The KEGG pathway analysis of the OTUs/ASV was inferred using Tax4Fun [34] (version 1.0). The FAPROTAX database (Functional Annotation of Prokaryotic Taxa) and associated software [35] (version 1.0) were used for generating the ecological functional profiles of bacteria. The functional group (guild) of the fungi was inferred using FUNGuild [36] (version 1.0). Redundancy analysis (RDA) was executed in the R project vegan package [37] (version 2.5.3) to clarify the influence of environmental factors on community composition. The Pearson correlation coefficient between ecological factors and species was calculated in the R project psych package [31] (version 1.8.4). A heatmap and network of correlation coefficients were generated using Omicsmart, a dynamic, real-time interactive online platform for data analysis (http://www.omicsmart.com), accessed on 20 April 2022. The datasets (SRP430770) presented in this study can be found in the NCBI Sequence Read Archive (https://www.ncbi.nlm.nih.gov/sra/?term=SRP430770), accessed on 4 April 2023).

## 3. Results

### 3.1. Effects of Different Composting Treatments on Soil TN, TP, TK, pH, and EC

Table 1 showed soil TN, TP, TK, pH, and EC changes under different treatments. Compared with the CK2 treatment, soil TN and TP contents were the highest in the T7 treatment, which increased by 9.38% and 10.47%, respectively, while soil TP increased by 5.39% in the T4 treatment, and there were no significant differences in soil TN and TP in the T2, T4, and T5 treatments. The soil TK content was significantly increased by 5.02% and 9.65% in the T4 and T7 treatments, respectively, compared to the CK2 treatment. Soil pH decreased from 0.01 to 0.11, with a slight decrease and no significant difference between treatments. In addition, T7 treatment significantly enhanced the soil EC.

### 3.2. Effect of Different Composting Treatments on Soil Available Nutrients and Organic Matter 

Table 2 lists the changes in soil availability of nutrients and organic matter under different composting treatments. Compared with the CK2 treatment, the T4 treatment significantly increased the contents of AN, AP, AK, and SOM by 6.16%, 6.92%, 15.66%, and 9.28%, respectively. The T7 treatment significantly increased the contents of AN, AK, and SOM by 8.80%, 14.97%, and 11.16%, respectively. These results suggested that the T4 and T7 treatments enhanced the soil’s availability of nutrients and organic matter more effectively.

### 3.3. Effect of Different Composting Treatments on Zucchini Yield

As shown in Figure 1, compared with the CK2 treatment, the yields of the T4 and T7 treatments were significantly higher than the local CK2 treatment, with increases of 23.1% and 18.0%, respectively. The remaining compost treatments were not significantly different, with the yield of each treatment showing T4 > T7 > T3 >T6 > T1 > CK2 > T5 > T2 > CK1. It showed that T4, T7, T3, T6, and T1 treatments were favorable to increasing the yield of greenhouse zucchini compared to local commercial organic fertilizer.

### 3.4. Effect of Different Composting Treatments on the Alpha Diversity of Bacterial and Fungal Communities in Soils

Figure 2 shows the α-diversity of soil bacterial and fungal communities in different compost treatments. The higher the values of the community richness indices (Chao1 and ACE), the higher the community richness. The higher the Shannon index, the higher the microbial diversity of the sample, and the higher the Simpson index, the lower the microbial diversity of the sample. Among the α-diversity of bacteria (Figure 2A), the ACE and Chao1 indices of T4 treatment increased by 1.88% and 2.07%, respectively, compared with CK2 treatment. In the alpha diversity of fungi (Figure 2B), the ACE and Chao1 indices of the T7 treatment decreased by 5.2% and 4.46%, respectively, compared with the CK2 treatment, and the Shannon index increased by 12.57%. The Simpson index increased by 5.75% compared with CK2. The above data showed the highest bacterial richness was found in the T4 treatment and the lowest fungal richness in the T7 treatment among the compost treatments. However, there was no significant difference overall. No significant change in bacterial diversity and a decrease in fungal diversity occurred after compost application.

### 3.5. Effect of Different Composting Treatments on Beta Diversity of Bacterial and Fungal Communities in Soils

The PCoA analysis showed that the different composting treatments altered the community composition of soil bacteria and fungi (Figure 3). For the bacterial community structure, the first and second principal coordinates explained 33.21% and 19.46% of the differences among the five treatments, respectively (Figure 3A). The bacterial communities of the CK2, T4, and T7 treatments were significantly separated from those of the CK1 treatment. Moreover, the CK2 and T7 treatments were closer, indicating that their bacterial community structure was more similar. For the fungal community structure, the first and second principal coordinates explained 18.79 and 14.51% of the variance, respectively (Figure 3B). The fungal communities of the four treatments were significantly separated. The differences in fungal communities between treatments were more pronounced.

### 3.6. Phylum-Level Composition and Relative Abundance of Soil Bacterial and Fungal Communities in Different Composting Treatments

The four treatments in this experiment were similar in microbial diversity but differed in abundance. The relative abundance of soil bacterial and fungal communities at the phylum taxonomic level for the four treatments is shown in Figure 4. The relative abundance of known bacterial taxa detected in the rhizosphere soil at the phylum level (Figure 1A) was *Proteobacteria* (23.97~28.34%), *Patescibacteria* (12.20~16.17%), *Bacteroidetes* (10.89~13.48%), *Acidobacteria* (7.99~9.08%), *Planctomycetes* (7.83~9.89%), *Chloroflexi* (6.95~7.72%), *Actinobacteria* (4.18~6.18%), *Gemmatimonadetes* (3.4~4.44%), *Verrucomicrobia* (3.14~4.10%), and *Firmicutes* (2.03~2.78%). The total sum of these bacterial populations ranged from 89.98% to 92.64%, which indicated that soil bacteria showed high diversity and abundance at the phylum level. The sum of the top six populations in terms of relative abundance in the T4 treatment was the highest at 79.35%, an increase of 3.2% over the CK2 treatment. Among the compost treatments, the relative abundance of Phylum *Bacteroidetes* and *Chloroflexi* increased by 2.69% and 0.99%, respectively, in the T4 treatment. The relative abundance of *Proteobacteria* and *Chloroflexi* increased by 1.98% and 1.25% in the T7 treatment compared with the CK2 treatment, and the *Patescibacteria* decreased to different degrees in both the T4 and T7 treatments. The bacterial phylum with the highest relative abundance in each compost treatment were *Acidobacteria* (9.08%) in the CK2 treatment; *Patescibacteria* (16.17%) and *Bacteroidetes* (13.48%) in the T4 treatment; and *Proteobacteria* (28.34%), *Chloroflexi* (7.72%), and *Actinobacteria* (5.06%) in the T7 treatment. It indicates that applying compost increases or reduces the abundance of dominant phyla in the rhizosphere of zucchini but does not affect the species of dominant phyla.

At the phylum level, the top 6 fungal taxa in relative abundance in the rhizosphere soil (Figure 4B) were *Anthophyta* (41.4%~45.3%), *Ascomycota* (25.34~39.68%), *Mortierellomycota* (4.80%~12.24%), *Nematoda* (2.31%~4.45%), *Chlorophyta* (1.94%~4.07%), and *Basidiomycota* (1.90%~4.18%). Among the compost treatments, the relative abundance of *Anthophyta* was reduced by 19.13% and 6.58% in T4 and T7 treatments compared to CK2 treatment, the relative abundance of *Ascomycota* and *Basidiomycota* was highest in T4 treatment, which increased by 14.34% and 2.23% compared to CK2, and the relative abundance of *Mortierellomycota* increased by 2.57% in T7 treatment compared to CK2 treatment. The fungal phylum with the highest relative abundance in each compost treatment were *Ascomycota* (39.68%), *Basidiomycota* (4.18%) in the T4 treatment, and *Mortierellomycota* (9.38%) in the T7 treatment.

### 3.7. Genus-Level Composition and Relative Abundance of Soil Bacterial and Fungal Communities in Different Composting Treatments

The variation in the relative abundance of soil bacterial and fungal communities at the generic level is shown in Figure 5, where the relative abundance of known bacterial taxa detected in rhizosphere soils at the generic level (Figure 5A) was *Sphingomonas* (1.84~2.75%), *Flavobacterium* (0.52~4.24%), *Nitrospira* (1.30~1.49%), *Steroidobacter* (1.27~1.51%), *Terrimonas* (1.24%~1.38%), *Ensifer* (0.48~2.00%), *Pseudomonas* (0.46~1.80%), *Pirellula* (1.05~1.39%), *Stenotrophobacter* (0.81~1.37%), *RB41* (0.87~1.30%), other (23.13~26.44%), and unclassified (58.91~64.59%). The total relative abundance of known bacterial taxa at the genus level ranged from 8.97% to 17.96%, with a negligible overall percentage. The T4 treatment had the highest total relative abundance of known bacterial taxa at the genus level (16.70%), an increase of 2.35% compared to the CK2 treatment. Among the compost treatments, the relative abundance of *Sphingomonas*, *Flavobacterium*, and *Pseudomonas* was the highest in the T4 treatment compared to the CK2 treatment, with increases of 0.48%, 2.77%, and 0.81%, respectively. The relative abundance of *Ensifer* was the highest in the T7 treatment, with a rise of 0.71%, followed by an increase of 0.53% in *Pseudomonas* in the T7 treatment—the relative abundance of *Stenotrophobacter*. The relative abundance of *Stenotrophobacte* and RB41 in T4 and T7 treatments decreased by 0.56%, 0.37%, 0.41%, and 0.17%, respectively, compared with the CK2 treatment. This indicates that the T4 treatment mainly increased the relative abundance of *Sphingomonas*, *Flavobacterium*, and *Pseudomonas*, and the T7 treatment increased the relative abundance of *Ensifer*, but the overall percentage of known bacterial taxa at the genus level was relatively small.

At the genus level, the known eukaryotic taxa detected in the rhizosphere soil (Figure 5B) were *Cucurbita* (32.31~51.79%), *Mortierella* (0.07~7.93%), *Zopfiella* (1.84~2.75%), *Pratylenchus* (1.84~2.38%), *Ascobolus* (1.40~2.73%), *Acutodesmus* (0.50~1.26%), *Fusarium* (0.35%~1.14%), *Cephaliophora* (0.41~1.58%), *Asterarcys* (0.23~1.00%), and *Aspergillus* (0.01%~2.00%). Among them, the *Cucurbita*, *Acutodesmus*, and *Asterarcys* belong to the Viridiplantae, and the *Pratylenchus* belongs to the Metazoa. The remaining six belong to the fungal boundary, of which the *Mortierella* belongs to the *Mortierellomycota* and the *Zopfiella*, *Ascobolus*, *Cephaliophora*, *Fusarium*, and *Aspergillus* all belong to the *Ascomycota*. After excluding *Viridiplantae* and *Metazoa*, the total relative abundance of known fungal taxa at the genus level ranged from 10.25% to 17.28%, with a relatively small overall proportion. The application of planting and breeding waste compost reduced the relative abundance of *Mortierella* and *Aspergillus* and elevated the relative abundance of *Zopfiella*, *Ascobolus*, and *Cephaliophora*. Among the compost treatments, the highest relative abundance of *Zopfiella* and *Fusarium* was found in the T4 treatment, with 7.94% and 1.15%, respectively. The highest relative abundances of *Mortierella*, *Ascobolus*, and *Cephaliophora* were found in the T7 treatment with 9.38%, 2.73%, and 1.57%, respectively, and these genera were elevated compared to CK2. The relative abundances of *Ascobolus* and *Cephaliophora* were 9.38%, 2.73%, and 1.57%, respectively. This indicates that T4 treatment mainly increased the relative abundance of *Zopfiella* and *Fusarium*, and T7 treatment mainly increased the relative abundance of *Mortierella*, *Ascobolus*, and *Cephaliophora*, but the overall percentage of known fungal taxa at the genus level was small.

### 3.8. Prediction of Soil Bacterial Tax4Fun Function after Application of Planting and Breeding Waste Compost

Tax4Fun is a KEGG (Kyoto Encyclopedia of Genes and Genomes Pathway) functional prediction based on the species annotation information from the Pathway database, i.e., the Metabolic Pathway database. The level1 of the metabolic pathway database classifies biological metabolic pathways into seven major categories: Metabolism, Genetic Information Processing, Environmental Information Processing, Cellular Processes, and Organismal Systems. Level2 can divide the 7 categories into 59 subcategories. From Figure 6, the top 20 subcategories were selected based on level 2 and belonged to 4 major categories. In terms of Metabolism, there were 11 subcategories. Among them, the bacterial community in the CK1 treatment had the highest predicted functional abundance in the Metabolism of Other Amino Acids, Metabolism of Cofactors and Vitamins, Glycan Biosynthesisand Metabolism; the bacterial community in the T4 treatment had the highest predicted functional abundance in Energy Metabolism, Nucleotide Metabolism, and the Metabolism of Other Amino Acids. T7-treated bacterial communities had the highest functional abundance in Lipid Metabolism, Xenobiotic Biodegradation and Metabolism. In Environmental Information Processing, there was one subcategory, among which the bacterial community treated with CK1 had the highest functional abundance in Signal Transduction. In Genetic Information Processing, there are three subcategories, among which the CK2-treated bacterial community has the highest functional abundance in Folding, Sorting, and Degradation, and T4 has the highest functional abundance in Translation. In Cellular Processes, there were three subcategories: Membrane Transport, Cell Growth, and Death, which were the most abundant in T4 treatment, and Cell Motility, which was the most abundant in CK2 treatment. The above results indicated that soil bacteria had the highest percentage and abundance of the Metabolism pathway. Applying planting and breeding waste composts significantly enhanced the abundance of Metabolism, Genetic Information Processing, and Cellular Processes in the soil bacteria.

### 3.9. Prediction of soil Fungal FUNGuild Function after Application of Compost to Planting and Breeding Waste

Information on the functional classification of the fungi in the samples and their abundance in each sample was obtained from the FUNGuild functional prediction (Figure 7). The results showed that the fungal community trophic types could be classified into seven categories: Pathotroph, Pathotroph-Saprotroph, Pathotroph-Saprotroph-Symbiotroph, Pathotroph-Symbiotroph, Saprotroph, Saprotroph-Symbiotroph, and Symbiotroph. The CK1, CK2, T4, and T7 treatments were all dominated by Pathotrophs (2.62%, 1.12%, 1.44%, and 0.20%), Pathotroph-Saprotroph-Symbiotrophs (9.37%, 5.59%, 10.37%, and 8.76%), Saprotrophs (4.26%, 5.33%, 12.98%, and 6.52%), Saprotroph-Symbiotrophs (14.58%, 7.55%, and 9.05%), and fungi (13.07%). Application of planting and breeding waste compost reduced the abundance of Pathotroph and Saprotroph-Symbiotroph fungi and elevated the abundance of saprotroph fungi. Compared with the CK2 treatment, the T4 treatment significantly increased the abundance of Pathotroph-Saprotroph-Symbiotroph and Saprotroph fungi, and the T7 treatment reduced Pathotroph and elevated the abundance of Pathotroph-Symbiotroph fungi.

The above seven trophic types were further classified, and 44 defined functional groups were detected by functional taxon identification, of which 20 functional groups (abundance >0.1%) significantly differed between treatments (Figure 8). Eight functional groups were significantly more abundant in the CK1 treatment than in the remaining treatments, namely Saprotroph [(Endophyte-Litter Saprotroph-Soil Saprotroph-Undefined Saprotroph)], Pathotroph [(Animal Pathogen)], Symbiotroph [(Ectomycorrhizal), (Endophyte)], Pathotroph-Saprotroph [(Plant Pathogen-Undefined Parasite-Undefined Saprotroph], (Plant Pathogen-Undefined Saprotroph), (Plant Pathogen-Undefined Saprotroph) (Plant Pathogen-Undefined Saprotroph)], Pathotroph-Saprotroph-Symbiotroph [(Animal Pathogen-Endophyte-Fungal Parasite-Plant Pathogen-Wood Saprotroph), (Ectomycorrhizal-Fungal Parasite-Plant Pathogen-Wood Saprotroph)]. The abundance of five functional groups was significantly greater in the T4 treatment than in the other treatments, and saprotrophs dominated these five functional groups [(Dung Saprotroph-Ectomycorrhizal-Soil Saprotroph-Wood Saprotroph), (Dung Saprotroph), (Plant Saprotroph-Wood Saprotroph)], followed by Saprotroph- Symbiotroph [(Bryophyte Parasite-Dung Saprotroph-Ectomycorrhizal-Fungal Parasite-Leaf Saprotroph-Plant Parasite-Undefined Saprotroph-Wood Saprotroph)], and Pathotroph-Saprotroph-Symbiotroph [(Animal Pathogen-Endophyte-Lichen Parasite-Plant Pathogen-Soil Saprotroph-Wood Saprotroph)]. Three functional groups were significantly more abundant in the T7 treatment than in the other treatments, and these three functional groups were dominated by saprotrophs [(Dung Saprotroph-Soil Saprotroph-Wood Saprotrop), (Dung Saprotroph-Soil Saprotroph)], followed by Pathotroph-Saprotroph-Symbiotroph [(Animal Endosymbiont-Animal Pathogen-Undefined Saprotroph)]. The above results indicate that applying planting and breeding waste compost significantly increased the abundance of Saprotroph and Saprotroph-Symbiotroph fungi and somewhat decreased the abundance of Pathotroph and Pathotrope-containing fungi.

### 3.10. Relationship between Soil Bacterial Community Structure and Soil Chemical Properties

The RDA analysis in Figure 9 showed the relationship between soil chemical properties and bacterial community composition. The first two axes of the RDA analysis explained 92.53% of the total variance in the soil bacterial community (axis 1: 75.73%; axis 2:16.80%) (Figure 9A). The pH showed the highest correlation with the soil bacterial community structure, followed by AK, AP, SOM, AN, TN, TP, TK, and EC. CK1 treatment was positively correlated with pH and negatively correlated with the rest of the indicators. On the contrary, CK2 and T4 treatments were negatively correlated with pH and positively correlated with the rest of the indicators. These environmental factors were significantly associated with bacterial community composition and are essential factors influencing bacterial community composition.

A correlation analysis of bacterial dominance phylum level flora with soil environmental factors (Figure 9B) showed that pH significantly correlated with *Gemmatimonadete*, *Elusimicrobia*, *Actinobacteria*, and *Omnitrophicaeota* and negatively correlated with *Hydrogenedentes*. *Proteobacteria* showed a significant positive correlation with AP and a highly significant positive correlation with the rest of the indicators. *Hydrogenedentes* showed a highly significant positive correlation with SOM, AP, and AK and a significant positive correlation with TN, TP, TK, and AN. In addition, SOM, EC, TN, TP, TK, AN, AP, and AK were positively correlated with *Bacteroidetes*, *Chloroflexi*, *Firmicutes*, *Cyanobacteria*, *Dependentiae*, and *Armatimonadetes*. A correlation analysis of the bacterial dominant genus level flora with soil environmental factors (Figure 9C) showed that *Flavobacterium* showed a significant positive correlation with SOM, AN, AP, and AK, a highly significant negative correlation with pH, and a positive correlation with the rest of the indicators. *Ensifer* showed a highly significant positive correlation with EC, a negative correlation with pH, and a significant positive correlation with the rest of the indicators. Pseudomonas showed a significant positive correlation with SOM, TN, AN, AP, and AK, a significant negative correlation with pH, and a positive correlation with the rest of the indicators. pH also showed a significant and highly significant positive correlation with *Pirellula* and *Stenotrophobacter*. In addition, *Sphingomonas* and *Steroidobacter* were positively correlated with SOM, EC, TN, TP, TK, AN, AP, and AK.

### 3.11. Relationship between Soil Fungal Community Structure and Soil Chemical Properties

The RDA analysis in Figure 10 shows the relationship between soil chemical properties and fungal community composition. The first two axes of the RDA analysis explained 95.29% of the total variance in the soil bacterial community (axis 1: 86.61%; axis 2: 8.68%) (Figure 10A). Soil chemical properties influenced the soil fungal community composition in the following order: pH > AP > AK > AN > TN > SOM > TP > TK > EC. The CK1 treatment was positively correlated with pH and negatively correlated with the rest of the indices. On the contrary, CK2, T4, and T7 treatments were negatively correlated with pH and positively correlated with the rest of the indices. The correlation analysis of fungal dominance phylum level flora and soil environmental factors showed (Figure 10B) that *Mucoromycota* showed a highly significant positive correlation with pH and a highly significant negative correlation with SOM, TN, TP, TK, AN, AP, and AK. *Mortierellomycota* and *Glomeromycota* showed a significant positive correlation with pH, and the rest of the indicators showed that the remaining indicators were negatively correlated to varying degrees. In addition, SOM, EC, TN, TP, TK, AN, AP, and AK were positively associated with *Ascomycota*, *Bryophyta*, and *Cercozoa*. The correlation analysis of fungal dominant genus level flora with soil environmental factors (Figure 10C) showed that *Mortierella*, *Acutodesmus*, *Asterarcys*, and Aspergillus showed positive correlations with pH to varying degrees and negative correlations with the remaining indicators. *Zopfiella*, *Ascobolus*, and *Cephaliophora* were positively correlated with indicators other than pH.

## 4. Discussion

### 4.1. Effect of Planting and Breeding Compost on Soil Chemical Properties and Zucchini Yield

Fertilizer application is essential to maintaining sustainable production on farmland. The reasonable application of planting and breeding compost is conducive to soil fertilization, improving soil physical and chemical properties, and providing rich nutrients and a good soil environment for crop growth and development [38]. In our study, we found that adding corn straw alone or mushroom residue and corn straw to sheep manure and tailing vegetables significantly increased the content of AN and AK in the soil, while the increased ratio of corn straw increased organic matter. The application of planting and breeding waste compost brought a large amount of organic matter into the soil, and the decomposition of organic matter enhanced the activity of microorganisms and enzymes related to nutrient transformation, thus increasing the effective nutrients of the soil [39]. On the other hand, straw with sheep manure and tailing vegetables can provide soil microorganisms with suitable carbon and nitrogen nutrients, thus maintaining and increasing microbial activity and numbers, accelerating the decomposition of organic matter and the conversion of mineral nutrients, and thus increasing the effective soil nitrogen content [40].

It has been shown that the application of organic fertilizer can not only increase the organic matter content of greenhouse soil, improve the soil’s buffering capacity against acid, and raise the soil pH, but also that the high decomposition rate of organic fertilizer in greenhouse soil increases the effective nutrients of the soil, improves the soil structure, promotes the growth of beneficial soil microorganisms, and inhibits the occurrence of vegetable diseases. Applying organic fertilizers also increases soil pH and helps alleviate soil acidification [41]. This experiment also found that soil pH decreased to different degrees after fertilization, but the decrease was minimal, probably because the application of planting and breeding waste compost and chemical fertilizers in combination coordinated the nutrients and inhibited the acidification tendency of the soil. In addition, the planting and breeding waste compost could effectively increase the soil organic matter content and alleviate soil slumping and acidification. This is mainly because the humus in organic matter can improve the soil pore condition and water and gas ratio, loosen the soil, and improve the soil structure [42]. The increase of AP content in soil may be due to the fact that when compost is applied to the soil, the increase of organic matter leads to the production of organic acids, which activate the phosphorus element in the soil and provide the soil with a large number of anions that form stable chelates with metal cations such as iron and aluminum, thus reducing the fixation of inorganic phosphorus and increasing the effectiveness of the phosphorus element [42,43].

Lignocellulose-rich corn stalks have improved the biological efficiency and yield of *Pleurotus eryngii* and *Pleurotus ostreatus* cultivation when added to the soil [44]. Sheep manure compost combined with chemical fertilizers can achieve high crop yields to a certain extent [45]. In our study, we also found that applying planting and breeding waste compost increased zucchini yield to varying degrees for the same amount of fertilizer. The addition of corn straw to compost significantly increased zucchini yield compared to commercial organic fertilizer, a result similar to the study of Wei et al. [46]. From the compost formulations, the T4 and T7 treatment formulations increased the SOM and AN content with no significant changes in other nutrients compared to the CK2 treatment formulation, indicating that the addition of corn straw or corn straw and the mushroom residue was beneficial to the compost nutrient enhancement and thus yield improvement. From the T2, T3, and formulation based on SM:TV = 6:3, the addition of straw or mushroom residue was beneficial to enhance the AN and SOM contents, with straw having the most significant effect, while all the AN, SOM, AP, AK, and TK contents showed different degrees of decline after the addition of cow manure. Replacing 10% of the tail vegetables with a mushroom resin based on T4 (SM:TV:CS = 6:3:1) increased the content of TN, AN, and AK and decreased the content of TK, while replacing 10% of the tail vegetables with corn straw on T3 (SM:TV:MR = 6:3:1) significantly increased the content of all nutrients except TK. It indicated that the effect of different wastes on yield at the same quality was in the order of corn straw > mushroom residue > cow manure; this is similar to the findings of Chen et al. [47].

### 4.2. Effect of Planting and Breeding Compost on Soil Microbial Diversity and Community Composition

Numerous studies have shown that the long-term application of chemical fertilizers alone can lead to a decrease in the abundance and diversity of soil bacteria and fungi, while the application of organic fertilizers alone or in combination with organic fertilizers can increase the abundance and diversity of soil bacteria and fungi [48,49]. In this study, soil without fertilizer application had the highest abundance of fungi and bacteria, which might be because several follow-up applications of chemical fertilizer in the zucchini cultivation system caused a decrease in the abundance of soil bacteria and fungi. Among the compost treatments, the T4 treatment enhanced bacterial richness, fungal richness, and diversity compared to the CK2 and T7 treatments, but there was no significant change in bacterial diversity, indicating that the fungal community was more sensitive to the compost response. Suzuki et al. [50] had concluded that fungal communities were more sensitive to inorganic and organic fertilizers than bacterial communities, which was consistent with our results. In contrast, the growth of the bacterial community was more closely related to soil type than fertilization regimes [50]. Further searching for species that differed significantly between sites at the phylum and genus levels revealed that, although multiple fertilizer applications led to a reduction in bacterial and fungal richness, compost maintained bacterial and fungal diversity and differential species numbers. Among them, the T4 treatment enhanced the fungal diversity, indicating that the compost formulation with high straw content enriched the soil fungal diversity. The increase in soil fungal diversity facilitated the decomposition of macromolecular organic matter, participation in nutrients, cycling, and prevention of soil-borne pests and diseases, etc. [51]. High-straw content compost provides more carbon and various nutrients for soil microorganisms, which is conducive to the growth and reproduction of microorganisms and their activity [52].

Different fertilization practices lead to the enrichment of specific bacteria or fungi that can efficiently use these nutrients, thus changing the composition of the microbial community. Elucidating soil microbial taxa at the phylum level can reveal the ecological coherence of microbial communities [53]. In this study, the dominant bacterial phyla in the rhizosphere of zucchini were: *Proteobacteria*, *Patescibacteria*, *Bacteroidetes*, *Acidobacteria*, *Planctomycetes*, *Chloroflexi*, and *Actinobacteria*. The application of planting and breeding waste compost mainly increased the relative abundance of *Proteobacteria* and *Bacterodetes* and decreased the relative abundance of *Planctomycetes* and the phylum *Actinobacteria*. From the composition of composting materials, adding mushroom residue and corn straw on top of sheep manure and tailing vegetables was conducive to increasing the relative abundance of the *Proteobacteria* and decreasing the relative abundance of the *Patescibacte*. In contrast, the addition of corn straw was conducive to increasing the relative abundance of Proteobacteria and *Bacteroidetes*, which have a solid ability to decompose macromolecular organic matter in the soil, especially *Proteobacteria* [54,55]. Mushroom residues were more difficult to decompose than corn straw, which resulted in a higher abundance of Proteobacteria in the T7 treatment. In contrast, the T4 treatment had the highest corn straw content, which provided sufficient carbon sources for the growth and reproduction of soil microorganisms, resulting in the highest total relative abundance of the dominant phylum in the T4 treatment. *Proteobacteria* are a group of functional microorganisms with nitrogen fixed. The increased abundance of *Proteobacteria* facilitates the accumulation, conversion, and utilization of nitrogen in soil [56]. The highest nitrogen content of compost in the T7 treatment caused the highest abundance of *Proteobacteria*. *Actinobacteria* are adapted to live in environments with low organic matter content. The input of organic matter in fertilizers led to a decrease in the abundance of *Actinobacteria*. However, *Bacteroidetes* are suitable to live in environments with high nutrient content, so their abundance showed an increasing trend [57]. The relative abundance of *Patescibacteria* was the highest in the T4 treatment because members of this group prefer environments with high carbon-to-nitrogen ratios and make full use of organic matter for secondary reproduction and growth during the composting process and after the compost is applied to the soil. The small genome of *Patescibacteria* is presumed to have a symbiotic or parasitic lifestyle. The higher abundance of *Patescibacteria* means that a large number of other microorganisms are available for their attachment [58,59]. This also resulted in the highest total relative abundance of dominant phyla in the T4 treatment.

Bacteria play a major role in the early stages of macromolecular organic matter degradation, and fungi play a major role in the later stages of degradation [60]. Fungi are important decomposers in soil, with strong decomposition ability and the ability to absorb and transform nitrogen from plant residues and hard-to-degrade organic matter such as cellulose, hemicellulose, and lignin in soil [61]. The application of planting and breeding waste compost mainly increased the relative abundance of fungi in the *Ascomycota* and *Basidiomycota*. It decreased not only the relative abundance of fungi in the *Mortierellomycota* but also the relative abundance of *Nematod*. *Ascomycota* in the T4 treatment was the absolute dominant phylum with a relative abundance of 39.68%, which increased by 14.34% compared with CK2. *Ascomycetes* degrade decaying organic substrates and are the main decomposers of soil organic matter (cellulose, lignin, and pectin), which is susceptible to plant species and residual straw [62]. The increase in the relative abundance of *Ascomycota* suggested an increase in organic matter and fertility in the soil [63]. In previous studies, the relative abundance of *Ascomycota* was found to increase significantly after the addition of straw [64]. In our study, the pH in treated soil was between 8~8.5, which is suitable for the growth of saprophytic fungi, and this explained why *Ascomycota* is the dominant phylum [65].

### 4.3. Effect of Planting and Breeding Compost on Soil Microbial Community Function

A total of 4 primary and 20 secondary functions were identified to predict bacterial Tax4Fun function. The results found that applied planting and breeding waste compost significantly enhanced the abundance of Metabolism, Genetic Information Processing, and Cellular Processes in the soil bacteria. In terms of Metabolism, the bacterial community of the T4 treatment had the highest functional abundance at Energy Metabolism, Nucleotide Metabolism, and Metabolism of Other Amino Acids, whereas the community of the T7 treatment had the highest functional abundance at Lipid Metabolism, Xenobiotics Biodegradation and Metabolism. In terms of Genetic Information Processing, the CK2 treatment had the highest functional abundance at Folding, Sorting, and Degradation, while T4 had the highest functional abundance at Translation. In terms of Cellular Processes, the highest functional abundance was found in Membrane Transport, Cell Growth, and Death for the T4 treatment. According to known reports, enhanced metabolic functions help to promote the decomposition of added materials and the accumulation of organic matter by bacterial communities [66]. Enhanced lipid metabolism can provide better energy to bacterial communities, and improved genetic information processing capacity leads to enhanced gene expression [67]. From a functional gene perspective, the addition of more straw to the compost increased the relative abundance of Energy Metabolism and Amino Acid Metabolism, which is consistent with a previous study that found that exogenous organic matter (e.g., leaf litter) enhanced the Amino Acid Metabolism capacity of soil microorganisms [68]. These changes in bacterial taxa and functional genes may drive nutrient cycling in the soil, which in turn promotes plant growth. The application of compost significantly enhanced the abundance of saprotroph fungi, with the highest in the T4 treatment at 12.98%, an increase of 8.72% and 7.65% compared to CK1 and CK2, respectively, indicating that planting and breeding waste compost enhanced the decomposition capacity of soil fungi. Some studies have shown that the increase of effective state N content in soil is positively influenced by soil saprophytic nutrient fungi [69,70]. Increasing the number of saprophytic fungi in soil can promote soil organic matter degradation, improve soil nutrient content and nutrient utilization, and reduce the risk of soil-borne diseases [71]. This may be one of the reasons for the higher AN and SOM content of the T4 treatment.

### 4.4. Relationship between the Microbial Community and Soil Environment Factors

The soil environment plays a critical role in influencing the structure of microbial communities, as evidenced by numerous studies. For example, Jin et al. [72] observed that AP, TN, SOM, EC, and pH were the primary drivers of microbial community structure, which is consistent with the results of our study. In the present study, RDA analysis revealed that the main factors that led to differences in communities between treatments were pH, AP, AK, and SOM, with pH varying in a small range but still significantly affecting the soil microbial community. The pH can directly affect microorganisms in the soil, and strong habitat pressure will retain those species that are more adapted to this pH [73]. It is also possible that pH greatly influences the growth and reproduction of microorganisms because it generally affects physiological and biochemical activities such as enzyme formation, enzyme activity, metabolic pathways, or cell membrane permeability, which in turn affects the diversity of the entire community. The reason for the significant effect of AP on soil microbial communities may be that soil microorganisms obtain sufficient carbon sources in the short term and multiply rapidly, during which they assimilate large amounts of phosphorus (mainly Olsen P). On the other hand, exogenous carbon sources increase the activity of soil phosphatase, which accelerates the decomposition of organic phosphorus and phosphorus conversion by microorganisms [74]. Soil microorganisms preferentially use phosphorus released from organic materials. Composts with high phosphorus content are more likely to decompose and release phosphorus when soil nitrogen is sufficient, and phosphorus is a limiting factor in the decomposition of materials with high carbon-to-nitrogen ratios [75]. *Proteobacteria* and *Bacteroidetes* showed a highly significant positive correlation with various soil environmental factors, such as SOM, EC, TN, TP, TK, AN, AP, AK, etc. Those results suggested that *Proteobacteria* and *Bacteroidetes*, as the two most dominant bacterial phyla, played a positive role in the degradation of organic matter and were more suitable for survival in soils with relatively rich nutrient contents. Likewise, *Ascomycota* and *Bacteroidetes* showed highly significant or significant positive correlations with a variety of soil environmental factors except for pH, and as the absolute dominant soil fungal phylum, their elevated relative abundance also indicated an increase in soil fertility [76,77,78]. A previous study pointed out that SOM, pH, and AP in soil were the main factors causing variation in soil fungal community structure and diversity [79]. In our study, the enhancement of fungal diversity in the T4 treatment was also associated with these environmental factors. In terms of compost formulation, the addition of corn straw had a more significant effect on soil environmental factors, and the addition of straw was also found to change the community structure of fungi in previous short-term culture trials and long-term field trials on corn straw return [53,80]. The T4 treatment in this experiment had the most significant effect on the soil environment, mainly altering SOM, AP, and AK in the soil. Therefore, it is presumed that compost formulations containing high amounts of corn straw returned to the field changed the soil’s physicochemical properties, which in turn affected the microbial community’s structure and diversity.

## 5. Conclusions

Planting and breeding waste compost can effectively reduce agricultural surface pollution and improve soil productivity, which is essentially important for sustainable agriculture. In the present study, we revealed that SM:TV:CM = 6:3:1 was the optimal formulation, which mainly enhanced the SOM, AP, and AK content in the soil and alleviated soil acidification. Although multiple fertilizer applications led to a decrease in bacterial and fungal richness, the compost maintained bacterial diversity and enhanced fungal diversity, and these changes eventually enhanced zucchini yield. This study has explored the localized compost technology of planting and breeding waste, and the results would provide a scientific basis for effectively promoting sustainable agricultural development.

## Figures and Tables

**Figure 1 microorganisms-11-01026-f001:**
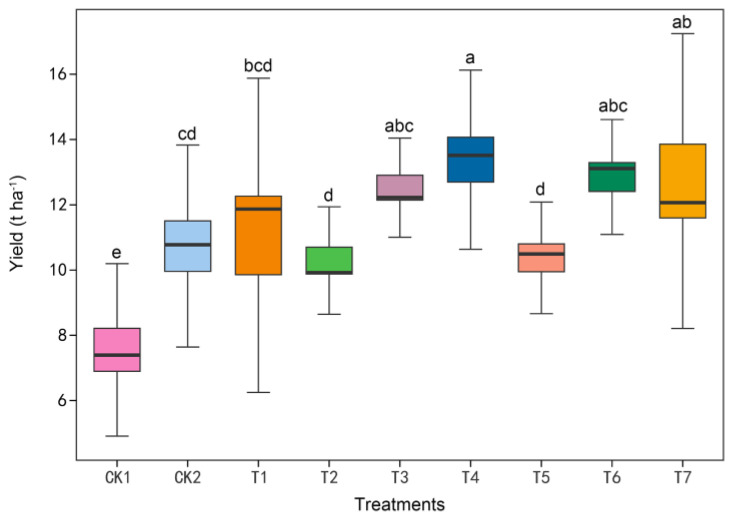
Yield of zucchini grown in soils with different composting treatments. The values presented are the means ± SE (*n* ≥ 5). Vertical bars indicate the SE of the means. Different lowercase letters simultaneously indicate significant differences among treatments (Duncan’s test, *p* < 0.05).

**Figure 2 microorganisms-11-01026-f002:**
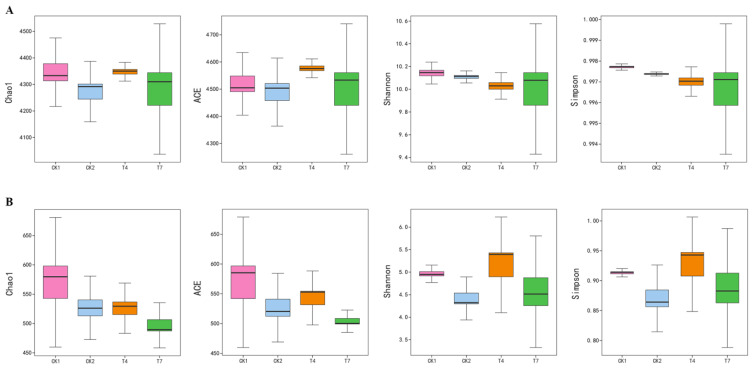
Effect of different composting treatments on the α-diversity of soil microbial communities. (**A**) α-diversity of the bacterial communities in rhizosphere soil. (**B**) α-diversity of the fungal communities in rhizosphere soil.

**Figure 3 microorganisms-11-01026-f003:**
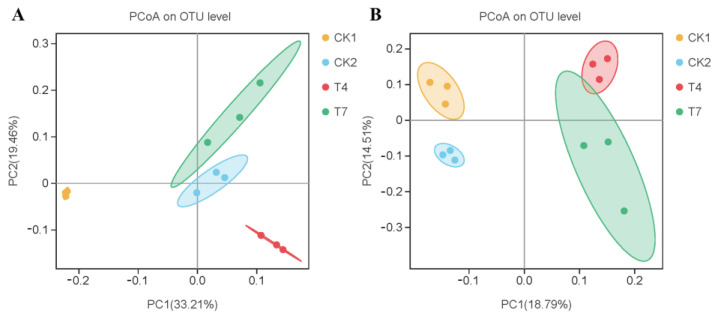
Principal coordinate analysis (PCoA) plots of bacterial (**A**) and fungal (**B**) community composition at the OTU level.

**Figure 4 microorganisms-11-01026-f004:**
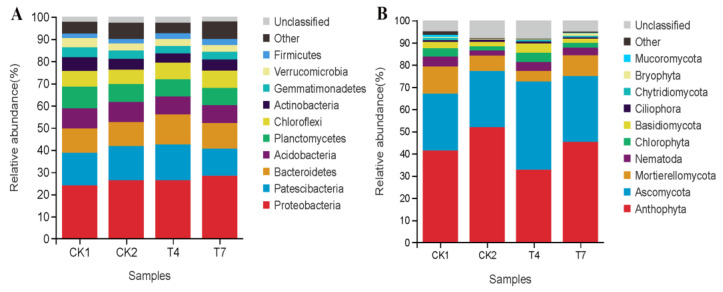
Composition and relative abundances of bacterial (**A**) and fungal (**B**) taxa at the phylum level.

**Figure 5 microorganisms-11-01026-f005:**
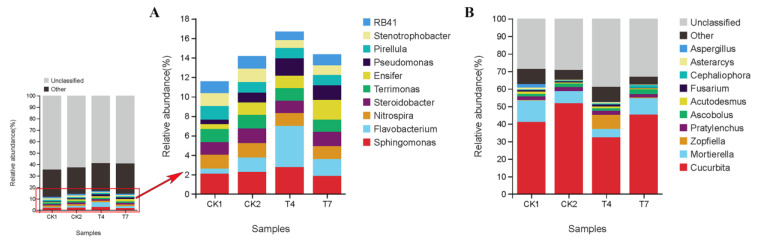
Composition and relative abundances of bacterial (**A**) and fungal (**B**) taxa at the genus level.

**Figure 6 microorganisms-11-01026-f006:**
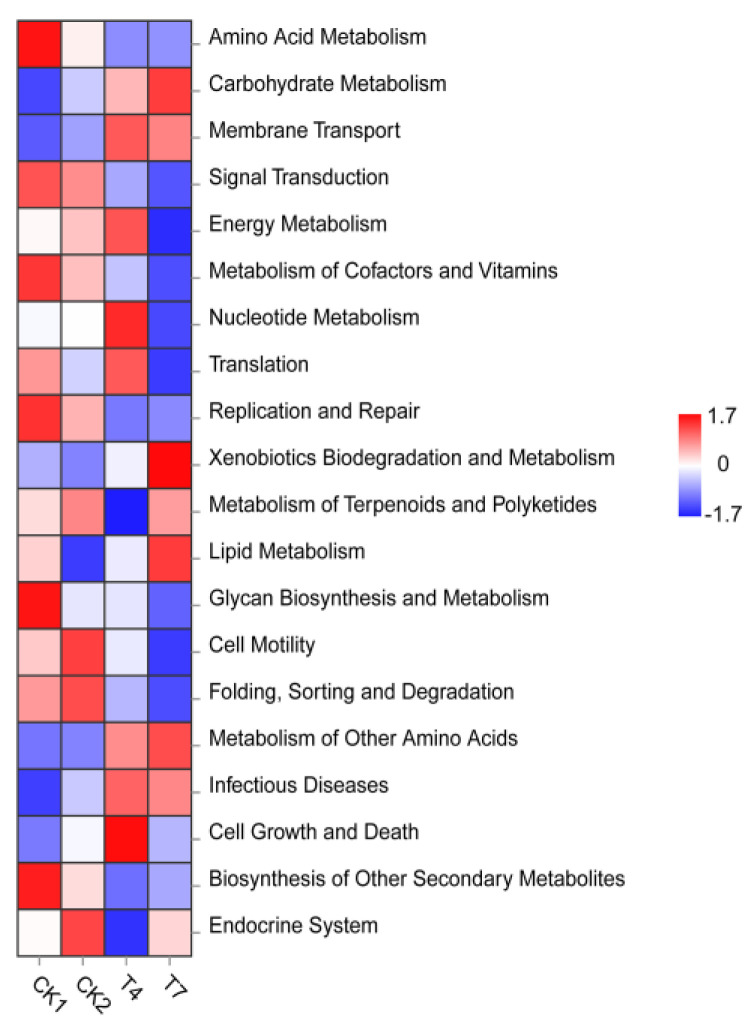
A heat map for predicting the functional contribution of bacteria in soils with different treatments based on Tax4Fun (Functional Category 2). (Colors from blue to red represent the relative abundance of the functions from low to high).

**Figure 7 microorganisms-11-01026-f007:**
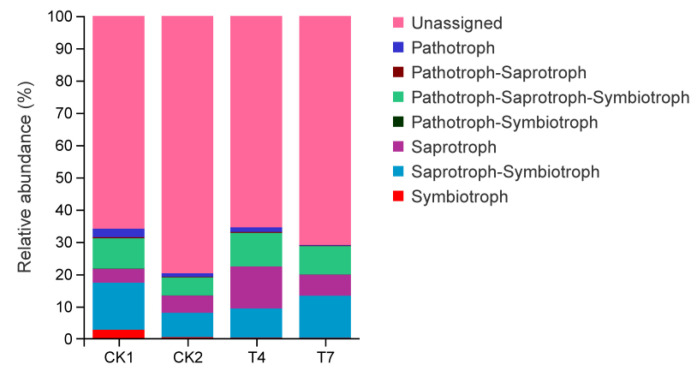
Prediction of fungal functions in different treated soils based on FUNGuild.

**Figure 8 microorganisms-11-01026-f008:**
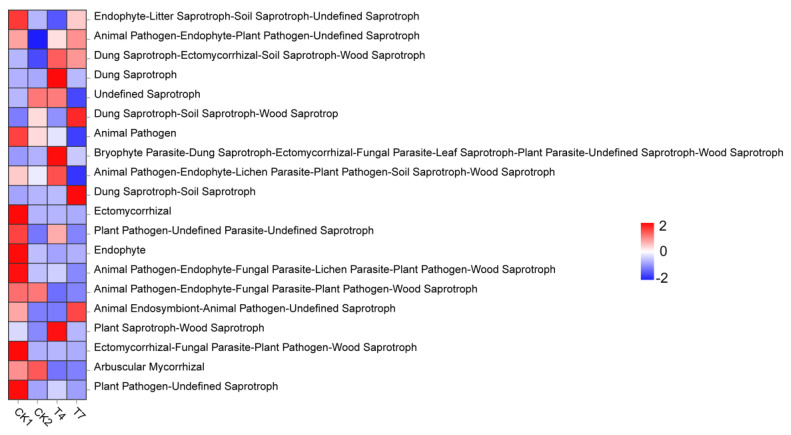
A heat map for predicting the functional contribution of fungi in soils with different treatments based on FUNGuild. (Colors from blue to red represent the relative abundance of the functions from low to high).

**Figure 9 microorganisms-11-01026-f009:**
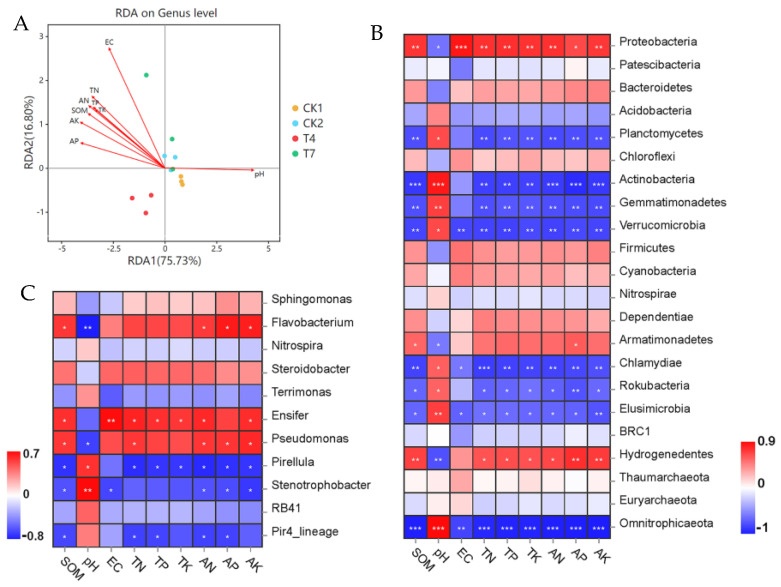
Redundancy analysis (RDA) of soil chemical properties and bacterial communities. (**A**) Phylum level (**B**) and genus level (**C**) bacterial dominant flora and soil environmental factors: Pearson correlation analysis; SOM, soil organic matter; EC, electrical conductivity; TN, total nitrogen; TP, total phosphorus; TK, total potassium; AN, hydrolyzable nitrogen; AP, available phosphorus; AK, available potassium. (The horizontal axis indicates environmental factors, the vertical axis indicates species, and colors from blue to red represent the relative abundance of the functions from low to high; *: significant correlation, *p* < 0.05; **: highly significant correlation, 0.001 < *p* < 0.01; ***: highly significant correlation, *p* < 0.001).

**Figure 10 microorganisms-11-01026-f010:**
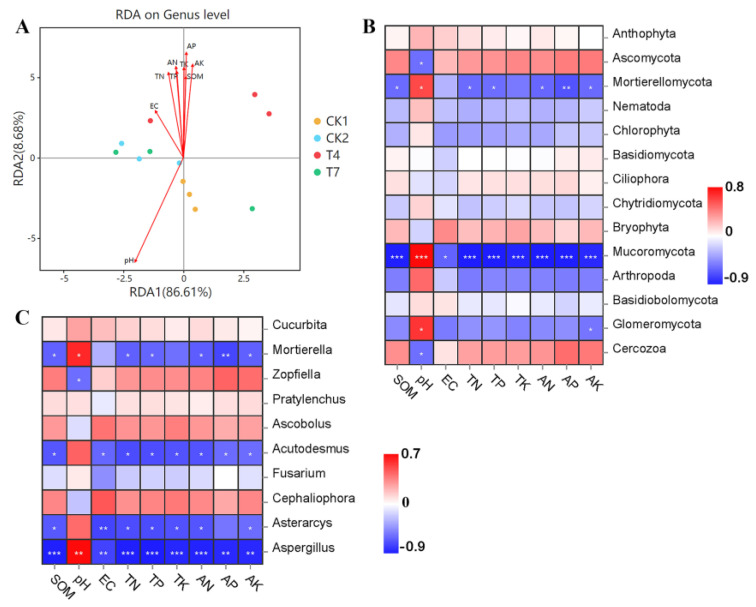
Redundancy analysis (RDA) of soil chemical properties and bacterial communities. (**A**) Phylum level (**B**) and genus level (**C**) bacterial dominant flora and soil environmental factors: Pearson correlation analysis; SOM, soil organic matter; EC, electrical conductivity; TN, total nitrogen; TP, total phosphorus; TK, total potassium; AN, hydrolyzable nitrogen; AP, available phosphorus; AK, available potassium. (The horizontal axis indicates environmental factors, the vertical axis indicates species, and colors from blue to red represent the relative abundance of the functions from low to high; *: significant correlation, *p* < 0.05; **: highly significant correlation, 0.001 < *p* < 0.01; ***: highly significant correlation, *p* < 0.001).

**Table 1 microorganisms-11-01026-t001:** Effects of different composting treatments on soil TN, TP, TK, pH, and EC.

Treatments	pH	EC (µS·cm^−1^)	TN (g·kg^−1^)	TP (g·kg^−1^)	TK (g·kg^−1^)
ck1	8.14 ± 0.0033 ^a^	305.97 ± 4.17 ^d^	0.18 ± 0.003 ^f^	2.37 ± 0.02 ^g^	16.58 ± 0.15 ^f^
ck2	8.09 ± 0.0067 ^b^	322.50 ± 1.32 ^bc^	0.32 ± 0.001 ^bc^	5.16 ± 0.10 ^c^	23.12 ± 0.09 ^c^
T1	8.09 ± 0.0133 ^b^	315.33 ± 1.20 ^cd^	0.22 ± 0.006 ^e^	3.46 ± 0.02 ^f^	23.71 ± 0.19 ^bc^
T2	8.11 ± 0.0033 ^b^	312.07 ± 0.64 ^d^	0.32 ± 0.003 ^c^	5.10 ± 0.09 ^c^	22.23 ± 0.29 ^d^
T3	8.10 ± 0.0067 ^b^	291.40 ± 2.08 ^e^	0.29 ± 0.001 ^d^	4.22 ± 0.07 ^d^	19.71 ± 0.07 ^e^
T4	8.04 ± 0.0058 ^c^	330.67 ± 4.10 ^b^	0.34 ± 0.003 ^b^	5.42 ± 0.03 ^b^	24.28 ± 0.04 ^b^
T5	8.10 ± 0.0058 ^b^	348.33 ± 4.41 ^a^	0.33 ± 0.005 ^b^	5.07 ± 0.13 ^c^	23.75 ± 0.46 ^bc^
T6	8.09 ± 0.0058 ^b^	324.33 ± 5.21 ^bc^	0.30 ± 0.005 ^d^	3.81 ± 0.15 ^e^	23.22 ± 0.16 ^c^
T7	8.06 ± 0.0058 ^c^	350.33 ± 1.45 ^a^	0.35 ± 0.004 ^a^	5.70 ± 0.05 ^a^	25.35 ± 0.42 ^a^

Values indicate a mean ± SE (*n* = 3). Different superscript letters in the same column represent significant differences among fertilizer treatments according to a one-way ANOVA (Duncan’s test, *p* < 0.05). EC, electrical conductivity; TN: total nitrogen; TP, total phosphorus; TK, total potassium.

**Table 2 microorganisms-11-01026-t002:** Effects of different composting treatments on soil types AN, AP, AK, and SOM.

Treatment	AN (mg·kg^−1^)	AP (mg·kg^−1^)	AK (mg·kg^−1^)	SOM (g·kg^−1^)
CK1	52.73 ± 0.82 ^h^	58.95 ± 0.64 ^h^	164.40 ± 3.09 ^g^	4.25 ± 0.083 ^f^
CK2	88.90 ± 0.73 ^d^	81.54 ± 0.59 ^c^	259.23 ± 0.67 ^d^	8.51 ± 0.050 ^c^
T1	86.33 ± 0.23 ^e^	74.00 ± 0.66 ^f^	230.03 ± 0.59 ^f^	8.06 ± 0.083 ^d^
T2	75.13 ± 0.31 ^g^	71.33 ± 0.28 ^g^	228.90 ± 1.30 ^f^	7.87 ± 0.050 ^e^
T3	91.82 ± 0.51 ^c^	73.26 ± 0.15 ^f^	288.70 ± 0.91 ^b^	7.91 ± 0.033 ^de^
T4	94.38 ± 0.51 ^b^	87.18 ± 0.50 ^a^	299.83 ± 3.85 ^a^	9.30 ± 0.019 ^a^
T5	82.95 ± 0.40 ^f^	78.30 ± 0.39 ^d^	243.00 ± 0.99 ^e^	8.38 ± 0.083 ^c^
T6	83.65 ± 0.20 ^f^	75.60 ± 0.37 ^e^	262.93 ± 0.38 ^d^	8.78 ± 0.038 ^b^
T7	96.72 ± 0.31 ^a^	82.98 ± 0.20 ^b^	298.03 ± 0.86 ^a^	9.46 ± 0.068 ^a^

Values indicate a mean ± SE (*n* = 3). Different superscript letters in the columns represent significant differences among fertilizer treatments according to a one-way ANOVA (Duncan’s test, *p* < 0.05). AN, hydrolyzable nitrogen; AP, available phosphorus; AK, available potassium; SOM, soil organic matter.

## Data Availability

The datasets (SRP430770) presented in this study can be found in the NCBI Sequence Read Archive (https://www.ncbi.nlm.nih.gov/sra/?term=SRP430770). Data not included within the manuscript are available upon written request to the corresponding author.

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
