# Peer review of "Yield and Rhizosphere Soil Environment of Greenhouse Zucchini in Response to Different Planting and Breeding Waste Composts"

_microorganisms, 2023, doi:10.3390/microorganisms11041026_

Round 1

Reviewer 1 Report

Thank you for giving me the opportunity to review the manuscript by Jianzhong Tie et al., entitledYield and rhizosphere soil environment of greenhouse zucchini in respond to different planting and breeding waste composts", submitted for publication in Journal of Microorganisms.

The presented work deals with To investigate the yield and rhizosphere soil environment of greenhouse zucchini in response to various planting and breeding waste compost, eight formulations were created for compost fermentation using agricultural waste [sheep manure (SM), tail vegetable (TV), cow manure (CM), mushroom residue (MR), and corn straw (CS)] without fertilizer (CK1) and local commercial organic fertilizer (CK2) as controls.

The manuscript is generally good written; however, I observed some minor grammar and syntax errors, as well as capitalization and punctuation errors throughout the manuscript text.

The article is characterized by modernity and detail in explaining the data. But it is excellent in terms of processing data, explaining results, giving scientific reasons and using good references Therefore, I recommend publishing after the language modification

In the following I provide numerous detailed comments, critiques, concerns and suggestions that should be considered before a final decision on the manuscript should be made. Considering my below-given critiques I believe that the revised manuscript will result in a very different version compared with its current state. Therefore, I suggest a re-submission of this work since it generally provides some interesting outcomes.

Sincerely

The main criticism points are:

1-      There are many grammatical, punctuation, syntax errors, so sever English language editing is needed. For example:

-       Line #90 – use (,) after enzymes instead of( .).

-       Line #95 – use use (,) after activities instead of ( .).

-       Line #95 – use (soil nutrients ) not (the soil nutrient )

-       Line #102 – write   combination based on yield

-       Line #109 – (respond) not (responds )

-       Line #124 – (of the north west ) not (of north west  )

-        There are many spelling errors Please review and amend it .

Author Response

请参阅附件

Reviewer 2 Report

Comment sheet

The research article entitled ‘Yield and rhizosphere soil environment of greenhouse zucchini in respond to different planting and breeding waste composts’ withholds importance in the field of study, rhizospsheric soil environment. This study seems to have equally importance to know the crop and rhizospheric microbes’ interaction. I would recommend this research article to accept after minor revision and concerns stated below.

1.     How many soil rhizospheric samples were used to analyze. In addition, authors need to provide study area more clearly.

2.     I do not clearly understand ‘breeding waste’ mentioned in this study. Please prove detail insights regarding this in introduction section.

3.     I couldn’t see the SRA (sequence read archive) accession numbers for the samples.

4.     This study seems about interactions of rhizospheric microbes and different cultivation practices, authors need to identify ‘key stone microbes’ in species level in each treatment they have followed.

Comment sheet

The research article entitled ‘Yield and rhizosphere soil environment of greenhouse zucchini in respond to different planting and breeding waste composts’ withholds importance in the field of study, rhizospsheric soil environment. This study seems to have equally importance to know the crop and rhizospheric microbes’ interaction. I would recommend this research article to accept after minor revision and concerns stated below.

1.     How many soil rhizospheric samples were used to analyze. In addition, authors need to provide study area more clearly.

2.     I do not clearly understand ‘breeding waste’ mentioned in this study. Please prove detail insights regarding this in introduction section.

3.     I couldn’t see the SRA (sequence read archive) accession numbers for the samples.

4.     This study seems about interactions of rhizospheric microbes and different cultivation practices, authors need to identify ‘key stone microbes’ in species level in each treatment they have followed.
